# Research on the management of the system construction of National parks with China characteristics: Evidence from policy texts

Jun-Hui Li[1], Hai-Tao Yu[2]*, Wen-Lie Chen[3], Yuan Wang[4]

1 School of Tourism Data/Artificial Intelligence, Guilin Tourism University, Guilin, China, 2 College of Tourism and Landscape Architecture, Guilin University of Technology, Guilin, China, 3 College of Economics and Management, Qinghai Minzu University, Xining, China, 4 School of Tourism Management, Guilin Tourism University, Guilin, China

* albertyht@glut.edu.cn

## Abstract

Quantitative evaluation of China's national park policy texts is an important window to understand its national park governance logic, top-level design of ecological protection, and future practice. This study focuses on three key areas: the central government's top-level design of national parks, local government responses, and the heterogeneity of these responses. By analyzing 77 national park policies in China, the following conclusions are drawn: (1) The central government's vision for the national park system emphasizes ecological conservation and sustainable livelihoods for indigenous communities. (2) There are significant differences between local governments in the specific implementation of national park policies. (3) The first five established national parks show significant differences in terms of conservation objectives, integrated management practices, spatial planning, industrial development directions. Its findings offer valuable inspiration for developing countries to establish national parks protection systems and provide direct reference for global biodiversity governance. And it will also play a key role in achieving the action-oriented global goals of Kunming-Montreal Global Biodiversity Framework.

## 1. Introduction

Since the establishment of Yellowstone National Park in 1872, recognized as the world's first national park [1], the concepts and practices surrounding national parks have steadily expanded across the globe. As of 2022, more than 4,000 national parks exist worldwide [2]. In China, the development of national parks began relatively recently, following a proposal at the Third Plenary Session of the 18th CPC Central Committee in 2013 [3]. Since then, state and local governments have introduced various policies to build a unified, well-regulated, and efficient national park system. National park policies in China are primarily aimed at preserving the originality and

**Data availability statement:** All relevant data are within the paper and its Supporting information files.

**Funding:** This research was funded by the National Social Science Foundation of China, grant number 20XSH022, and by the Philosophy and Social Science Planning key Project of Qinghai Province, grant number 23ZCZD007, and by the Philosophy and Social Science Foundation of Guangxi Zhuang Autonomous Region, grant number 21GMZ009.

**Competing interests:** The authors have declared that no competing interests exist.

integrity of natural ecosystems, ensuring collective ownership, and fostering a generational legacy [4]. The scale of China's national park system is remarkable; by 2035, with 49 national parks covering 10.3% of the country's land area.

China's approach to national parks is distinctive in terms of its historical background, current status, and governance models. First, the concept of national parks in China emerged during a critical transition towards ecological awareness [5]. These parks also serve as habitats for numerous indigenous communities [6], such as the 67,400 residents living within the Giant Panda National Park. This contrasts significantly with the vast, uninhabited wilderness areas commonly found in North American national parks. Globally, national park governance generally falls into two categories: centralized public management, as in the U.S. and Canada [7], and regional management, as practiced in Japan and South Korea [8], where conservation lands are not exclusively state-owned. China adopts a hybrid model, where governance is shared between the National Forestry and Grassland Administration and provincial governments.

These complexities complicate the direct application of existing research, necessitating independent analysis to discern both the alignment and variation between national and local directives relevant to Chinese national parks. A close examination of China's national park policy texts may reveal both consistency and divergence between central and local governance levels. Such insights can contribute to the sustainable development of national parks in China while also providing valuable references for national park management strategies in other countries.

The unique characteristics of China's national park system have attracted considerable scholarly attention. Policy evaluation, in particular, has been the focus of extensive research, aiming to provide insights for policymakers to refine their strategies [9]. For instance, Wang & Zeng. (2020) employed the Policy Modeling Consistency (PMC) index method to quantitatively compare and analyze 14 representative national park policies in Hainan, Singapore, and Hong Kong [10]. Zhu et al. (2024) use the multiple streams framework as a theoretical and through the lens of policy entrepreneurs, to furnish a renewed look at the policymaking process of conservation policy in China [11]. Chai et al. (2025) proposes a triadic "Natural-Social-Digital" framework for assessing China's national park policy effects [12]. Javeed et al. (2024) examined the impact of national park policies on biodiversity, ecological processes, water resource conservation, as well as research and education [13]. Zhang et al. (2023) explored shifts in authority and inter-departmental dynamics during the national park policy formulation process, applying theories of bureaucratic politics and actor-centered power [14].

Research on national park policy evaluation has primarily focused on policy outcomes, with limited attention given to quantitative analyses of policy texts, particularly in the Chinese context. And no prior studies quantitatively evaluated national park policy texts in China using natural language processing (NLP) and grounded theory. Additionally, there is a lack of studies examining the mechanisms through which local governments respond to central directives in this domain. Most policy evaluation methodologies are qualitative or rely on the traditional Policy Modeling Consistency

(PMC) index model, both of which have limitations due to potential biases in expert judgments and challenges in integrating diverse information for evaluating policy variables.

To address these gaps, this study investigates the mechanisms through which local governments in China respond to state-level national park directives. It examines the consistency and diversity of national park policies at both central and local levels, as well as the variations among the policy texts of the first five national parks. This work makes four key contributions. First, it is the first quantitative evaluation of Chinese national park policy texts using NLP and grounded theory. Second, it constructs a policy evaluation dictionary and introduces a novel quantitative evaluation index system, serving as a foundation for future policy assessments. Third, it proposes a mixed-method approach that integrates NLP, grounded theory, and statistical analysis. Finally, it provides evidence through the deconstruction and comparative analysis of policy texts, which may support the further refinement of China's national park strategies. Thus, this paper not only establishes a practical basis for the adjustment and optimization of China's national park policies but also introduces innovative techniques for the quantitative evaluation of national park policies globally.

The remainder of this paper is organized as follows: Section 2 presents a literature review that establishes the relationship between national park policy and policy evaluation. Section 3 outlines the policy documents analyzed in this study and details the methodological framework employed to evaluate policy responses. Section 4 reports the findings of the evaluation, focusing on the central government's top-level design for national parks in China, local government responses to this vision, and the heterogeneity of policy texts across the first five national parks. Section 5 discusses the implications of the findings and provides a summary. Section 6 addresses the limitations of the study and suggests directions for future research.

## 2. Literature review

### 2.1 Research on National park policy

National park policy represents a complex governance system that balances multiple objectives and stakeholders. While existing research has produced significant findings, it has primarily focused on three themes:

2.1.1 **The evolution of national park policy instruments.** Initial research examined the design and implementation of individual policy instruments. For example, in terms of legal regulation, Canada's national park legal system provides an appropriate model, as its comprehensive nature encompasses by-laws including but not limited to wildlife, endangered species, hunting, fireproofing, traffic, architecture, and community involvement [15]. National park policy practices in China illustrated how top-down design could catalyze institutional transformation [16]. The success of performance management in protected areas of countries of young democracy largely depends on the network flattening and real involvement of locals and indigenous people in public governance [17]. Brazil's national park laws similarly stipulate, the effective participation of local populations in the creation, implementation, and management of national park [18]. Regarding economic instruments, studies have investigated fee policies [19], eco-compensation [20] and tourism promotion [21] as mechanisms connecting national park ecological value with public awareness. These investigations increasingly recognized that, as Kubo et al. (2019) [22] observed, strategically combining policy instruments is essential for enhancing national park policy effectiveness. However, existing research has not established a systematic framework to evaluate the interaction of different policy instruments in complex governance situations. Especially for a country as vast as China, it remains unclear how to optimize the tool mix to coordinate central goals and local diversity in national park development. This theoretical gap prompts this article to adopt a research method that combines NLP and grounded theory.

2.1.2 **The challenges in national park governance models.** A fundamental challenge in policy implementation concerns governance structures, specifically the distribution of responsibilities between central and local authorities. There are three types of relations between the local government, National Park Service (NPS), and communities: those oriented by the NPS, by local government, or by a mixed type [23]. Multiple case studies indicate that achieving equilibrium between centralization and decentralization remains a global challenge. In contrast to the USA where all

national parks are managed by the federal National Park Service following the same management policies [24]. However, in a centralized system, decision-making takes place at a high level and far away, which, among other things, makes it difficult for local people to exchange relevant information with the manager of the PA and vice versa [25]. Organization and governance in the national park region are influenced by local administration, politics and traditional institutions [26]. Functional zoning [27,28], as a crucial management strategy, represents a spatial manifestation of this powers and responsibility allocation. While these studies have identified central-local tensions, they mainly describe implementation outcomes or summarize single-country models, rather than analyzing how such tensions are addressed in policy formulation. China's distinctive hybrid system, combing centralized unity with hierarchical management, is characterized by central policy texts that both accommodate and limit local variations [29], particularly regarding the exploration of consistency and heterogeneity between central and local levels in the context of China.

2.1.3  **The synergy of national park policy objectives.**  Achieving mutual benefits for ecological conservation and community development represents the optimal policy objective. Successful examples like Nanshan National Park demonstrate how benefit-sharing mechanisms can empower local communities [30], while failed cases highlight the consequences of community exclusion. Consequently, scholars advocate for integrated approaches including collaborative management and ecological compensation [31,32]. Currently, a new approach classifies community participation in conservation into three categories: "Community-involved Management", "Community Co-management", and "Community-led Management" [33]. While existing research emphasizes community participation, it often treats it as a uniform concept. Critical operational questions regarding participation methods decision-making involvement, and benefit distribution remain insufficiently analyzed through detailed policy-text examination.

## 2.2  Policy evaluation based on policy texts

2.2.1  **Single-policy evaluation methodologies.**  Policy evaluation is a multidimensional analysis that leverages existing data, theoretical frameworks, and modeling techniques to examine the origins and impacts of policies [34]. Currently, policy evaluation methods are generally classified as either qualitative or quantitative. Qualitative methods include text mining [35], photographic observation [36] and the Delphi method [37], among others. Quantitative evaluation methods include choice experimentation [38], generalized synthetic control [39], data envelopment analysis [40], difference-in-difference (DID) [41], Regression Discontinuity Design (RDD) [42], Propensity Score Matching (PSM), or Synthetic Control Method. Recent advances in NLP have led to major breakthroughs in analyzing policy texts. This technological progress is especially impactful in the analysis of national park policy documents. It enables researchers, practitioners, and policymakers to process large amounts of textual data, facilitating real-time information access, improving decision-making, and boosting governance efficiency [43]. While NLP has shown significant benefits in policy text analysis, its use in quantitative policy research still requires more investigation [44]. Notably, there is a lack of quantitative studies using NLP to examine Chinese national park policy texts.

2.2.2  **PMC index modelling.**  Within existing policy evaluation frameworks, the PMC Index has emerged as a scientifically validated and operationally robust method for assessing policy texts. This model has been applied to evaluate various types of policies, including battery recycling [45], carbon reduction [46], energy transition [47], cultivated land protection [48], and waste separation management [49]. As mentioned earlier, most previous studies use PMC-Index, which has significant limitations in setting policy variables and weights. First, the evaluation results are highly dependent on the preset variable system and may not capture the unique regional concepts and governance logic in the policy. While, we used NLP to analyze 77 national park policies. At this stage, the aim is to naturally identify core policy clusters from large-scale text data, avoid preset framework bias, and ensure that research conclusions are rooted in the original texts.

Second, this method cannot reveal policy semantic strength, contextual relevance, and assumes that secondary variables are of equal importance and cannot achieve variable weight allocation [50]. Therefore, this study uses a systematic three-level process of open coding→main axis coding→selective coding to analyze the causal relationship, contextual

correlation and strategic logic between policy elements, focusing on examining the semantic strength, contextual correlation and deep conceptual connection of the policy. The indicator weights determined by combining qualitative and quantitative methods can more accurately reflect the influence of each indicator.

In the analysis process (see below for specific steps), macro-level NLP findings provide direction for micro-level grounded theory research, and the coding results of grounded theory can feed back to the NLP model for verification and optimization. For example, word vector models can be retrained based on new recognition concepts. This iterative process significantly improves the reliability and interpretability of research results [51].

## 3. Materials and methods

### 3.1 Data collection and pre-processing

The establishment of a national park system in China was first proposed in November 2013. Therefore, the national park policy texts selected for this analysis cover the period from 2013 to 2023. These documents were sourced from the Beijing University Laws and Regulations Database (http://www.pkulaw.cn/) and various government websites, such as the State Council in China (http://www.gov.cn), the National Park Administration (http://www.forestry.gov.cn). A search using the term "national park" yielded 424 policy texts. Filtered to exclude invalid text, text missing the issuing authority, document number or publication date. At the same time, meeting notices, duplicate policies, approval letters, and multi-batch catalog policies are eliminated. Additionally, less relevant documents were eliminated, such as bank announcements regarding specific national park coins and industry regulations covering the sale of such coins. Finally, 77 valid texts were retained in the corpus. Detailed analysis of these documents is omitted here due to space constraints.

As shown in Fig 1, the number of national park policy texts in China has fluctuated but increased overall since 2013. This development trajectory can be segmented into three distinct stages: the policy exploration stage (2013–2016), the

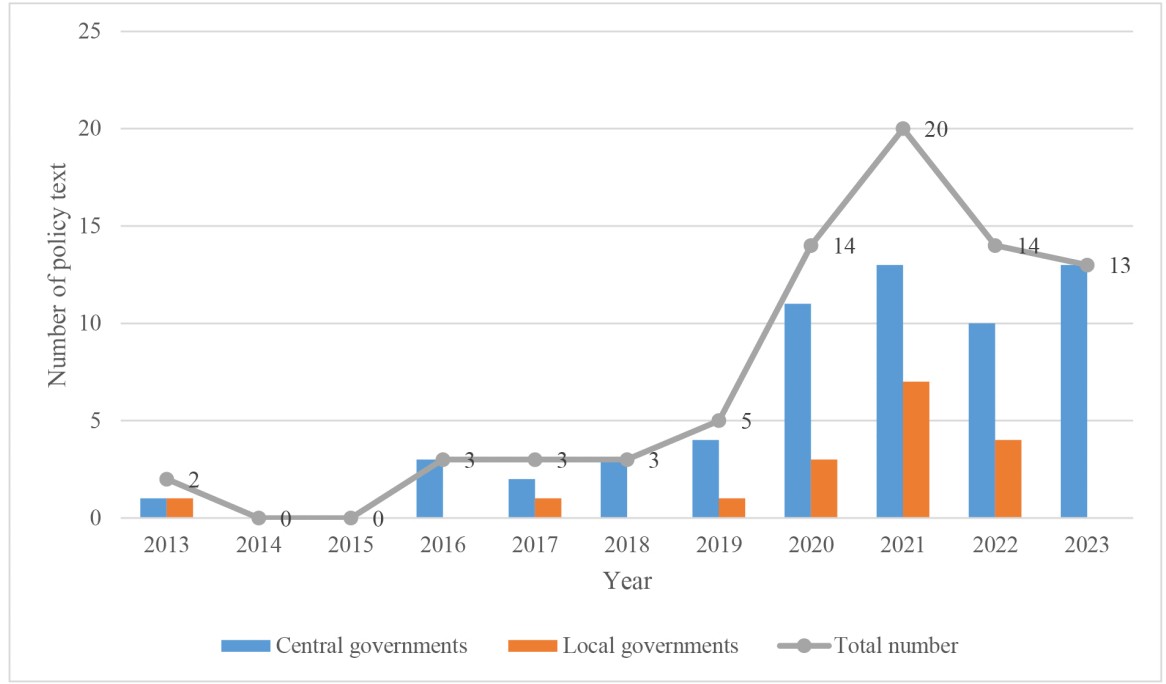

**Fig 1. The development of China's national park policies from 2013 to 2023.**

policy growth stage (2017–2020), and the policy maturity stage (2021–2023). During the policy exploration stage, there were only five national park policies concentrated in the provinces of Taiwan, Heilongjiang, and Yunnan. This number increased to 25 during the policy growth stage and surged to 47 during the policy maturity stage, when first five national parks were officially established, signifying the commencement of China's national park system construction.

## 3.2 Research design

The core methodology and key procedures utilized in this analysis are illustrated in Fig 2. In the first step, as shown in Fig 1, 77 national park policies were divided into two groups: central level (17 policies, comprising 22.08% of the total) and local level (60 policies, accounting for 77.92%). Subsequently, the policies were processed via word segmentation to remove irrelevant phrasing, add pertinent terminology, and create a specialized dictionary relevant to China's national park system using a Python script. A quantitative index system to evaluate the policies was then devised based on grounded theory techniques—open coding, axial coding, and selective coding—anchored in word frequency statistics and categorization. The subsequent step involved assigning weights to each index, merging both objective metrics and subjective judgments. A composite score for each policy was then determined. Separate evaluations of central and local national parks policies were conducted, facilitating a comparative analysis between the two tiers and examination of the heterogeneity across the policy texts of the first five national parks.

## 3.3 Policy correspondence evaluation method

**3.3.1 Dictionary construction, variable identification.** We used a Python program for word segmentation, exclusion of non-essential words, inclusion of specialized terms, and word-frequency tallying and sorting, for example, jieba.posseg for Part-of-speech tagging, jieba.analyse for Keyword Extraction, SnowNLP for natural language processing. The primary objective of this process was to construct a dictionary that encapsulates the characteristics of China's national park policies. The research team first conducted a pilot coding exercise to establish an overall coding framework. Subsequently, two research assistants independently coded 77 documents based on this framework. Coding reliability was tested using Cohen's kappa coefficient for consistency. After testing, it was found that the kappa coefficient of the two researchers coding the national park policy text was 0.812(>0.8). It can be seen that the coding results of this time are reliable (Table 1).

Grounded theory, a bottom-up approach proposed by Glaser and Strauss [52], served as the foundation for this analysis. This approach enables a rigorous, flexible, and in-depth theoretical inquiry that is based on empirical data, thus ensuring reliability and validity [53,54]. Three coding stages (open, axial, and selective) are systematically employed to construct a robust theoretical framework [55]. To maintain coding objectivity and precision throughout our process, four professional graduate students were arranged into two coding teams. Any instances of coding discrepancies were resolved by a third group comprised of our research team.

To maintain coding objectivity and precision throughout our process, four professional graduate students were arranged into two coding teams. Any instances of coding discrepancies were resolved by a third group comprised of our research team.

1) Open Coding

Open coding (Table 2) entails interpreting, continuously analyzing, refining, summarizing, and conceptualizing raw text data. Initially, policy text sentences were broken down and used to build a dictionary for conceptualizing and categorizing. Then, concepts with similar meanings were grouped for analysis, generating open coding categories. This process yielded 133 basic categories.

2) Axial Coding

Axial encoding (S1 Appendix) involves excavating deeper underlying relationships between the basic categories to establish higher-level main categories. A total of 49 basic variables were summarized by examining the connections among

1. Data Collection and Preprocessing
- Data Collection

Search for national park policies in China under the theme of "National Parks" from 2013 to 2023.

- Data Preprocessing

Eliminating those policies with low relevance and repetitive content.
Divide data into two types:
At the central level(17 policy texts)
At the local level(60 policy texts )

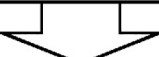

2. Construct the Dictionary and Variable Identification
- Construct the Dictionary

Word segmentation of national park policy texts
Removal of stop words
Adding self-defined words
Word frequency statistics and sorting

- Variable Identification

Open coding → Axial coding → Selective coding

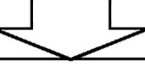

3. Weight Determination
- Subjectively calculating weights
- Objectively calculating weights
- Weighted average of both

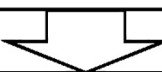

4. Policy Response Evaluation
- Top-level Design of National Parks
- Local Response of National Parks
- Comparison Between Top-level Design and Local Response
- Heterogeneity of Local Responses

**Fig 2. Research process and main steps.**

**Table 1. Classification standard-of Cohen's kappa coefficient.**

| Cohen's kappa coefficient | <0.2 | 0.21~0.40 | 0.41~0.60 | 0.61~0.80 | 0.81~1.00 |
|---|---|---|---|---|---|
| Level of consistency | Poor | Weak | Moderate | High | Excellent |

**Table 2. The process of coding the openness of national park policies (partial).**

| Serial Number | Original policy text | Basic category |
|---|---|---|
| P1 | **The purpose** of **establishing national parks** is to **protect the authenticity and integrity of the natural ecosystems**, always emphasizing **the strict**, **holistic and systematic protection of natural ecosystems**, and **the protection** of the areas that **should be protected the most**. | Ecological protection |
| P1 | **Integrate the management functions** of relevant **nature reserves**, in **conjunction** with **the reform of the management system of ecological environmental protection**, **the reform of the management system of natural resource assets**, and **the reform of the system of natural resource supervision**, the **management responsibilities** of **national parks** and **nature reserves** will **be exercised** by **a unified department**. | Constructing a unified management organization |
| P1 | **Rationalize the division of responsibilities** between **the central** and **local governments**, and **establish a coordinated management mechanism** for national parks with **clear subjects**, **clear responsibilities** and **mutual coordination**. | Clarify the responsibilities and powers of the central and local authorities |
| P1 | **Establishment** of a sound **volunteer service mechanism** and **social supervision mechanism**. | Volunteer service; Social supervision |
| … | … | … |
| P77 | … | … |

the 133 basic categories. These 49 basic variables were further categorized into seven major groups: policy basis, policy objectives, policy measures, policy areas, policy receptors, policy functions, and policy guarantees.

3) Selective Coding

Selective coding involves discovering core categories through clarifying "storylines", based on the findings from open and axial coding to clarify, sort, and integrate the relationships among concepts and variables. The core categories identified in this study constitute the proposed framework for quantitative analysis of China's national park policy texts, encompassing seven main variables.

The evolution of China's national park policy texts had unfolded over numerous phases. Initially, recognizing the need for building an ecological civilization, policymakers established several goals for establishing a national park system. Subsequent measures to foster the development of national parks were complemented by assurances to facilitate the smooth execution of these policies, which impacted beneficiaries in various regions. The policy functions both mirrored the foundational policy rationalize and furthered the realization of set goals. This cycle of development, as depicted in S2 Appendix, captures the dynamics of China's national park policy evolution.

4) Theoretical Saturation Test

We tested the concept of theoretical saturation with 10 previously reserved national park policy texts that had not been subjected to coding. No new concepts, categories, or interrelations emerged upon applying the three-tiered coding process to them, suggesting that the proposed model has a robust level of theoretical saturation. The model and coding outcomes were also reviewed by two national park field experts, whose affirmative feedback further confirmed the model's validity. Thus, the theoretical model shown in S2 Appendix effectively meets theoretical saturation test criteria.

**3.3.2 Weight determination.** Weight calculation methods can be roughly divided into subjective, objective, and combined approaches. The subjective method is susceptible to the inherent uncertainties of expert opinions, whereas the objective method may overlook the valuable insights of decision-makers [56]. Combined methods offer a workable balance, incorporating expert judgment while maintaining objectivity.

We adopted a combined weight determination method in this study. For the subjective component, we consulted 15 national park experts who provided individual assessments of the 49 secondary variables. The weights of these variables were computed from the experts' average ratings, which contributed to the primary variables/ weights. Objectively,

secondary variable entropy values were derived from keyword frequency, and word relevance was determined through semantic similarity analysis. The word entropy and keyword relevance informed a weighting function that established each variable's final weight [57]. These subjective and objective weights were arithmetically averaged to produce a composite weight for each variable (Table 3).

Among the primary variables, policy measures and policy basis occupy important positions, with weights of 0.247 and 0.234, respectively. The focus in terms of secondary variables is predominantly on realistic problems, predictions, national strategies, environmental justice, and NGO involvement, with weights of 0.128, 0.062, 0.055, 0.041, and 0.040. These results signify the considerable influence of these factors within the framework for evaluating China's national park policies.

**3.3.3 Policy response evaluation.** Based on the weight determination, we calculated the scores for each variable across the 77 policy texts using a linear summation model. The model yielded a comprehensive evaluation score for each policy after multiplying the variable weights by the processed data, followed by a simple summation.

$$Y_k = \sum_{j=1}^{n} w_j X_j$$

Where $Y_k$ is the comprehensive evaluation score of policy k, $w_j$ is the weight of policy k under variable j, $X_j$ is the normalized value of the word frequency of evaluation policy k under variable j, and n is the number of variables for the evaluation object.

As a core indicator in statistics, standard deviation is a statistical measure that gauges the dispersion of values in a series relative to the mean, serving as the root of the variance [58]. It intuitively reflects the variability and volatility of the data set: higher values indicate more significant data fluctuations, and lower values mean weaker volatility.

## 4. Results

### 4.1 Top-level design of National parks

Upon calculating the comprehensive scores, averages and corresponding percentages for each variable within the 17 national park policy texts were determined (Fig 3).

For primary variables, the average score of policy measures is the highest at 138.94, accounting for 54.37% of the total. The average scores for policy areas, policy functions, and policy guarantees are also significant at 38.62, 23.57, and 21.77, respectively, respectively accounting for 15.11%, 9.23%, and 8.51%. These figures suggest that the strategic framework of China's national parks adequately prioritizes these aspects.

Regarding secondary variables (S3 Appendix), the average scores for integrated management, environment field, planning and construction, industrial transformation and upgrading, organizational leadership, evaluation monitoring, specification, improving quality and efficiency, ecological protection and restoration, and realistic problems are relatively high at 62.24, 33.68, 27.70, 14.67, 13.73, 10.65, 8.55, 7.29, 5.53, and 5.13, accounting for 24.35%, 13.18%, 10.84%, 5.74%, 5.37%, 4.17%, 3.35%, 2.85%, 2.16%, and 2.01% of the total, respectively. These results demonstrate that the central government's vision for the construction of China's national park system echoes an ecological protection-first ethos and prioritizes the sustainable well-being of indigenous peoples, with an emphasis on harmonizing managerial agencies, standardizing criteria, enhancing monitoring processes, and fostering shared values.

### 4.2 Local responses to National parks

How the local government, as the implementation body of the central policy, responds to the top-level design in formulating the national park policies is an issue worth exploring. Based on calculating the comprehensive score, the average scores of each variables of the 60 national park policy texts at the local level are calculated, as well as their percentages, and the final results are shown in Fig 3.

**Table 3. The weights of variables in the quantitative evaluation system of national park policies.**

| Primary variables | Subjective Weight | Objective Weight | Combination Weight | Secondary variables | Subjective Weight | Objective Weight | Combination Weight |
|---|---|---|---|---|---|---|---|
| $X_1$ | 0.011 | 0.456 | 0.234 | $X_{1-1}$ | 0.003 | 0.030 | 0.016 |
| | | | | $X_{1-2}$ | 0.005 | 0.251 | 0.128 |
| | | | | $X_{1-3}$ | 0.002 | 0.067 | 0.034 |
| | | | | $X_{1-4}$ | 0.001 | 0.108 | 0.055 |
| $X_2$ | 0.099 | 0.127 | 0.113 | $X_{2-1}$ | 0.032 | 0.001 | 0.016 |
| | | | | $X_{2-2}$ | 0.011 | 0.005 | 0.008 |
| | | | | $X_{2-3}$ | 0.032 | 0.002 | 0.017 |
| | | | | $X_{2-4}$ | 0.017 | 0.018 | 0.017 |
| | | | | $X_{2-5}$ | 0.001 | 0.025 | 0.013 |
| | | | | $X_{2-6}$ | 0.006 | 0.076 | 0.041 |
| $X_3$ | 0.423 | 0.071 | 0.247 | $X_{3-1}$ | 0.023 | 0.002 | 0.013 |
| | | | | $X_{3-2}$ | 0.060 | 0.000 | 0.030 |
| | | | | $X_{3-3}$ | 0.021 | 0.004 | 0.013 |
| | | | | $X_{3-4}$ | 0.029 | 0.002 | 0.015 |
| | | | | $X_{3-5}$ | 0.066 | 0.000 | 0.033 |
| | | | | $X_{3-6}$ | 0.058 | 0.001 | 0.030 |
| | | | | $X_{3-7}$ | 0.014 | 0.001 | 0.008 |
| | | | | $X_{3-8}$ | 0.003 | 0.001 | 0.002 |
| | | | | $X_{3-9}$ | 0.042 | 0.002 | 0.022 |
| | | | | $X_{3-10}$ | 0.002 | 0.004 | 0.003 |
| | | | | $X_{3-11}$ | 0.021 | 0.004 | 0.013 |
| | | | | $X_{3-12}$ | 0.029 | 0.024 | 0.027 |
| | | | | $X_{3-13}$ | 0.040 | 0.022 | 0.031 |
| | | | | $X_{3-14}$ | 0.015 | 0.003 | 0.009 |
| $X_4$ | 0.094 | 0.120 | 0.107 | $X_{4-1}$ | 0.014 | 0.004 | 0.009 |
| | | | | $X_{4-2}$ | 0.021 | 0.060 | 0.040 |
| | | | | $X_{4-3}$ | 0.018 | 0.047 | 0.033 |
| | | | | $X_{4-4}$ | 0.026 | 0.002 | 0.014 |
| | | | | $X_{4-5}$ | 0.004 | 0.003 | 0.003 |
| | | | | $X_{4-6}$ | 0.010 | 0.005 | 0.008 |
| $X_5$ | 0.131 | 0.034 | 0.082 | $X_{5-1}$ | 0.016 | 0.022 | 0.019 |
| | | | | $X_{5-2}$ | 0.030 | 0.003 | 0.017 |
| | | | | $X_{5-3}$ | 0.007 | 0.005 | 0.006 |
| | | | | $X_{5-4}$ | 0.000 | 0.003 | 0.002 |
| | | | | $X_{5-5}$ | 0.077 | 0.000 | 0.039 |
| $X_6$ | 0.141 | 0.150 | 0.145 | $X_{6-1}$ | 0.014 | 0.110 | 0.062 |
| | | | | $X_{6-2}$ | 0.018 | 0.009 | 0.013 |
| | | | | $X_{6-3}$ | 0.007 | 0.009 | 0.008 |
| | | | | $X_{6-4}$ | 0.022 | 0.001 | 0.012 |
| | | | | $X_{6-5}$ | 0.019 | 0.012 | 0.016 |
| | | | | $X_{6-6}$ | 0.003 | 0.004 | 0.003 |
| | | | | $X_{6-7}$ | 0.020 | 0.004 | 0.012 |
| | | | | $X_{6-8}$ | 0.039 | 0.001 | 0.020 |

*(Continued)*

 

**Table 3.** (Continued)

| Primary variables | Subjective Weight | Objective Weight | Combination Weight | Secondary variables | Subjective Weight | Objective Weight | Combination Weight |
|---|---|---|---|---|---|---|---|
| $X_7$ | 0.102 | 0.043 | 0.072 | $X_{7-1}$ | 0.030 | 0.001 | 0.015 |
| | | | | $X_{7-2}$ | 0.026 | 0.003 | 0.015 |
| | | | | $X_{7-3}$ | 0.008 | 0.012 | 0.010 |
| | | | | $X_{7-4}$ | 0.003 | 0.014 | 0.008 |
| | | | | $X_{7-5}$ | 0.015 | 0.008 | 0.011 |
| | | | | $X_{7-6}$ | 0.020 | 0.005 | 0.012 |

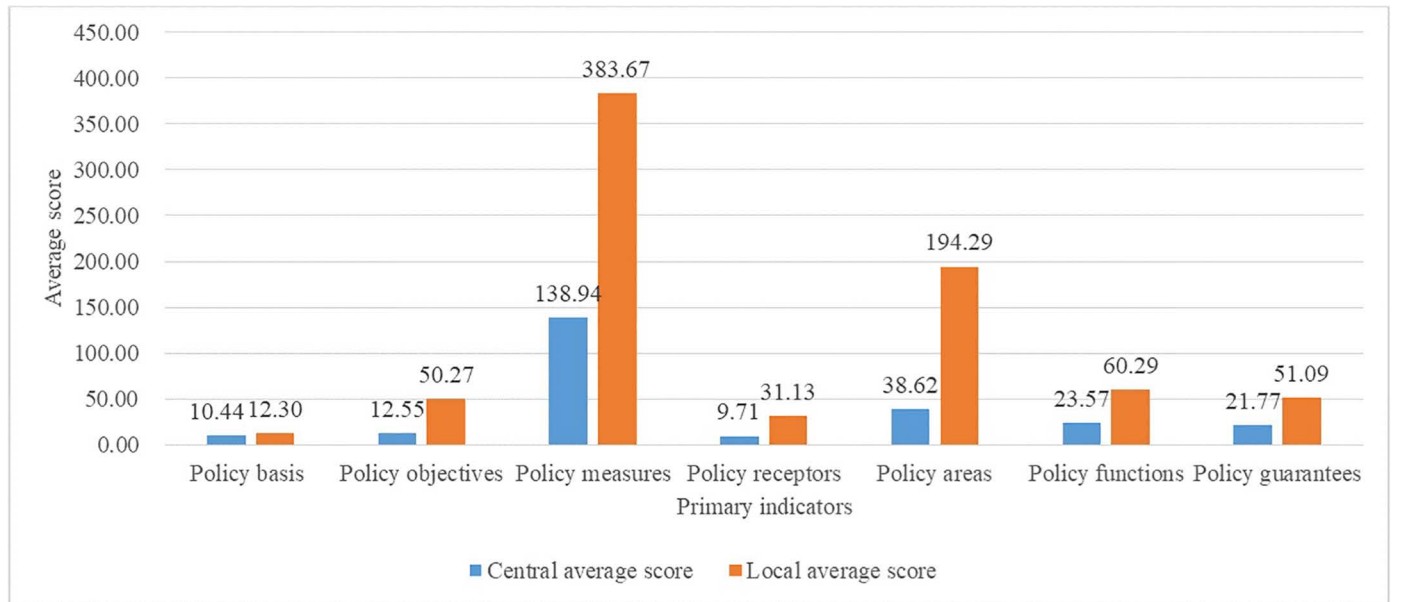

**Fig 3. The comparison of average scores for primary variables between the top-level design and local response.**

In terms of primary variables, the average score of policy measures is the highest with 383.67, accounting for 49.00%. And the average scores of policy areas, policy functions and policy guarantees are higher with 194.29, 60.29, and 51.09, accounting for 24.81%, 7.70%, and 6.52%, respectively. This is similar to the focus of the top-level design described above, implying that local national park policies are aligned with the focus of the central government's vision.

In terms of secondary variables (S3 Appendix), the average scores of environment field, integrated management, planning and construction, industrial transformation and upgrading, organizational leadership, improving quality and efficiency, specification, ecological protection and restoration, description, community residents are higher, with 177.22, 146.28, 90.94, 52.57, 28.60, 25.60, 25.43, 24.50, 14.47, and 14.39 accounting for 22.63%, 18.68%, 11.61%, 6.71%, 3.65%, 3.27%, 3.25%, 3.13%, 1.85%, and 1.84%, respectively. These proportions closely mirror those outlined in the central government's strategic design, suggesting close alignment between local and central policy objectives and measures in the realm of national park management.

### 4.3 Comparison between top-level design and local responses

**1) Comparison of primary variables.** A comparative visualization of the average scores for primary variables between the central government's strategic planning and local governments' implementation is provided in Fig 3. Local policies appear to be more granular and precise, as reflected by the significantly higher average scores.

The percentages of primary variables at both central and local levels are contrasted in Fig 4. There appears to be a mutual focus on policy measures and policy areas, which aligns with previous findings. However, discrepancies between top-level designs and responses from local governments are fairly pronounced in terms of policy areas, policy measures, and policy bases, with differences of 9.70%, 5.37%, and 2.51%, respectively—all higher than 2.0%.

**2) Comparison of secondary variables.** The average scores for secondary variables between central and local levels differ significantly, with standard deviations mostly exceeding 1, as shown in S3 Appendix. We observed a significant difference between central strategic design and local government implementation in this analysis. Variables like the ecological field, integrated management, planning and construction, and industrial transformation and upgrading exhibit particularly high standard deviations of 101.50, 59.42, 44.72, and 26.79, respectively.

To further probe the distinctions between central strategic designs for national parks and local government responses, we examined high-frequency words associated with the same variables at both levels. The results are presented in Table 4.

There are clear variances in ecological protection measures, integrated management behaviors, spatial planning expressions, and industrial transformation paths. In the ecological domain, central plans prioritize holistic ecosystem preservation and biodiversity maintenance, while local responses elaborate on the specific content of ecological protection measures. Central designs advocate for unified, centralized management, whereas local responses highlight institutional reforms and cross-regional collaboration. In the planning and construction of the national parks, central directives provide coordinates and spatial planning guidance; local responses materialize these into comprehensive, specialized plans and executable programs. Regarding industrial transformation and upgrading, top-level designs accentuate franchising operations, industrial transfer, and green industry initiatives, while local responses center on the concrete expansion of sectors

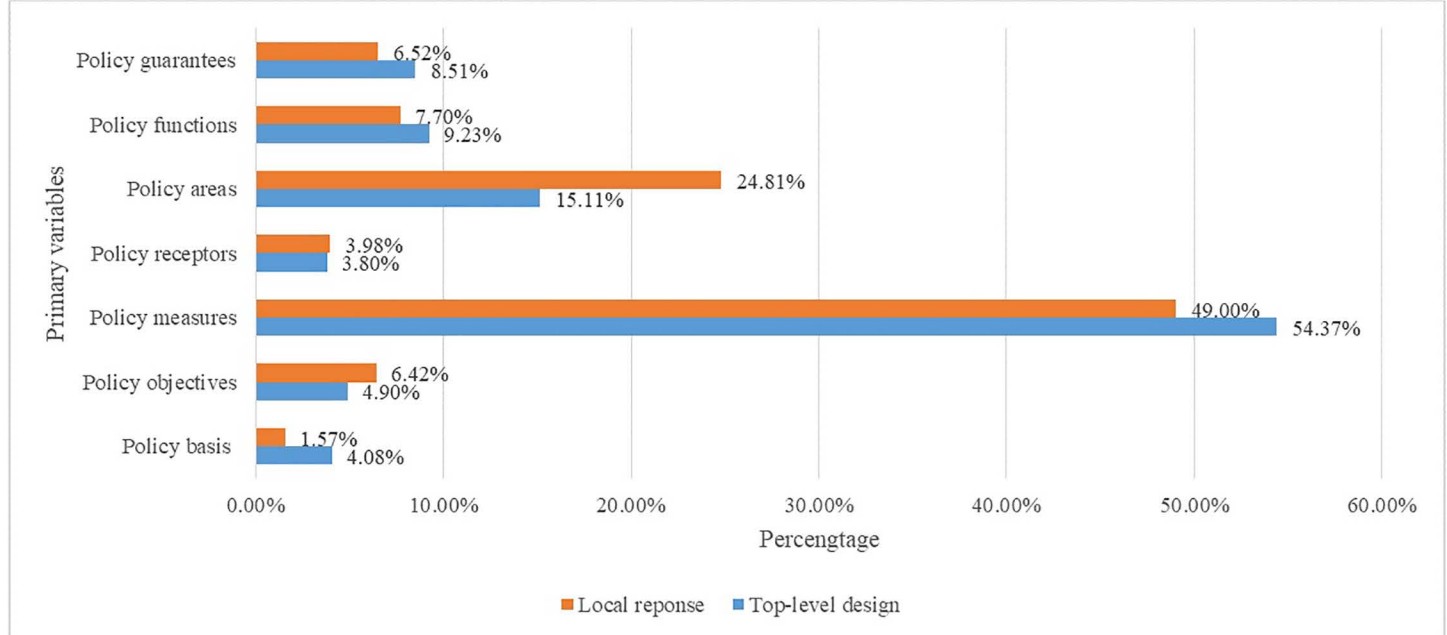

**Fig 4. Comparison of percentage of primary variables between the top-level design and local response.**

**Table 4. Comparison of high-frequency words contained in the same secondary variable at the central and local levels.**

| Secondary variables | Level | High frequency words |
|---|---|---|
| Ecological field | Central | Natural ecosystems, Nature reserves, Natural resources, Nature reserves, Biodiversity, Wildlife, Ecological civilization, |
| | Local | Tropical rainforest, Northeastern tiger, Northeastern leopard, Giant panda, Habitat, Wildlife, Biodiversity, Forest, Grassland, Wetland, Forestry |
| Integrated management | Central | Centralized, Unified leadership, Unified standards, Unified arrangements, Direct exercise, Universal ownership, National parks authority |
| | Local | Governing body, Location, Two provinces, Inter-provincial administration, Provincial, |
| Planning and construction | Central | Planning, Spatial planning, Construction projects, Rationalization, General layout |
| | Local | Master Plans, Special Plans, Specific Programs, Entrance communities, Ecological Corridors, Implementation Programs |
| Industrial transformation and upgrading | Central | Franchising, Industry transfer, Green industry, Eco-tourism, Vacation and leisure |
| | Local | Free-range farming, Breeding, Animal husbandry, Eco-tourism, Recreational experience, Nature experience, Cultural and creative products, Ecological science and technology, |

like livestock, eco-tourism, and cultural and creative industries. These results suggest that local governments are actively advancing national park development in harmony with the central government's overarching vision.

### 4.4 Heterogeneity of local responses

We selected the master plans for the first five national parks as a sample for this analysis. Details regarding these policies and their issuance periods are provided in S4 Appendix. We calculated the standard deviation of scores for the secondary variables to gauge the diversity of local national park policies across different regions; these findings are summarized in S5 Appendix.

The policies of the first five national parks differ greatly in terms of the ecological field, integrated management, planning and construction, and industrial transformation and upgrading, with standard deviations of 176.78, 115.03, 79.71, and 61.93, respectively. These figures notably exceed those for other secondary variables, illustrating the distinct localization of local policy responses.

We also analyze high-frequency terminology from these five national parks' master plans. As shown in Table 5, there are significant differences in ecological conservation targets, integrated management approaches, planning schemes, and industrial growth trajectories among the parks. While adhering to the central government's broader directives, local governments adapt in more nuanced ways as evident in the policy texts. These differentiated responses are potentially informed by each park's unique geographical features, climatic conditions, cultural heritage, and socio-economic context.

## 5. Discussion, implications, and findings

### 5.1 Discussion

Global discourse on protected area governance often presents a binary opposition between the perceived effectiveness of centralized control and the flexibility of decentralization [59]. Our analysis of China's national park policy documents, however, reveals a viable hybrid model. This model integrates mandatory ecological protection "red lines" set by the central government—which serve as non-negotiable baselines—with the flexibility for local governments to explore diverse implementation pathways above these thresholds, such as concession management and ecological compensation mechanisms. This "hard bottom line with flexible space" framework facilitates central-local collaborative governance, ensuring national strategic unity while stimulating local initiative, thereby offering valuable insights for global biodiversity conservation and ecological governance.

Balancing nature conservation with local socioeconomic development presents a global challenge, particularly in developing countries [60]. As the world's largest developing country, China's national park system offers relevant insights.

**Table 5. Comparison of high-frequency words included in the same secondary variable in the five national park master plans.**

| Secondary variables | Policy | High frequency words |
|---|---|---|
| Planning and construction | P7 | Northeast Tiger, Northeast Leopard, Wildlife, Habitat, Northeast Tigers and Leopards, Forests, Hoofed Species, etc. |
| | P9 | Natural resources, Habitat, Ice, Flora and fauna, Tibetan antelope, Yak |
| | P19 | Tropical rainforests, Ecosystems, Biodiversity, Nature reserves, Natural resources, Gibbons |
| | P23 | Flora and fauna, Habitat, Natural resources, Mixed forests, Broadleaf forests, Subtropical, Natural forests |
| | P26 | Giant panda, Habitat, Ecosystems, Wildlife, Forests |
| Ecological field | P7 | Northeast Tiger and Leopard National Park Authority, Two provinces, Russia |
| | P9 | Dynamic management and control, Sanjiangyuan National Management Organization, Dual leadership, Together |
| | P19 | Management body, Comprehensive management, Ownership, Rights registration, Synergistic management mechanism, Management bureau, Local government, Local conditions |
| | P23 | Wuyishan National Park Administration, Wuyishan City, Coordination, Local, Fujian Province |
| | P26 | Concurrence, Integration, Statistical table, Administrative district authority |
| Integrated management | P7 | Entrance communities, Core protection areas, Eco-camps, Control areas, Observatories |
| | P9 | Yangtze River source, Yellow River source, Lancang River source, General control area, Core protection area |
| | P19 | Test area, Sensitive area, Ecological corridor, Nature reserve, Ecological core area |
| | P23 | Protected areas, Monitoring stations, Aprons, Isolation zones, Ecosphere, Demonstration areas, Distribution maps, Warning signs |
| | P26 | Spatial planning, Protected areas, Signs, Master plan, Control areas, Ecological corridors |
| Industrial transformation and upgrading | P7 | Nature Recreation, Concession, Eco-tourism, Recreation Experience, State Forests |
| | P9 | Organic animal husbandry, Industry chain, Tourism, Ecological experience, Nature education |
| | P19 | Franchising, Green development, The Li nationality, The Miao nationality, Tourism |
| | P23 | Tea Culture, Science Education, Nature Education, Recreation Experience, Forest Carbon Sink, Digitalization, Industry Chain |
| | P26 | Franchise, Forestry, Culture, Eco-experience, Eco-tourism, Planting, Breeding |

Initiatives such as concession management, ecological compensation, and the "one household, one industry" program have effectively fostered local community participation. By leveraging their regional management strengths, local governments can further establish a collaborative governance framework that integrates administrative agencies, non-profit organizations, and community residents [61]. This approach helps optimize stakeholder participation and generate synergistic effects for conservation and development.

Although the national park policy framework provides a clear blueprint for central-local coordination, several implementation challenges warrant in-depth examination. It is important to note that our analysis, based on policy documents, inherently carries the limitation that written policies may not fully capture on-the-ground realities. First, a tension exists between policy uniformity and implementation diversity. While master plans employ similar structural frameworks, regional variations in land tenure, community composition, and economic development necessitate adaptive local practices, potentially leading to divergent outcomes from uniform policies. This gap between textual prescription and practical adaptation underscores a key limitation of purely document-based analysis. Second, entrenched sectoral path dependence may undermine the policy's "unified management" goal. Within the existing administrative structure, high costs of cross-departmental coordination can result in goal deviation or implementation inefficiencies, challenges that policy texts themselves may acknowledge but cannot fully resolve in practice. Finally, the effectiveness of community engagement requires continuous monitoring. The long-term sustainability of "shared benefits" outlined in policy texts depends on whether communities secure substantial and equitable economic returns and achieve meaningful representation through ecotourism and concession schemes. This highlights the critical distinction between the normative aspirations embedded in policy and their tangible realization.

## 5.2 Theoretical contributions

First, this study developed a dictionary for evaluating national park policies, along with an innovative quantitative evaluation index system. These tools were used to characterize the developmental trajectory of China's national park policies. As a result, this work advances the quantitative evaluation of national park policies, with particular emphasis on the interaction dynamics between central and local governance. The index system introduced here provides new perspectives on policy analysis that could inform future research on national park policies in other countries.

Second, the proposed hybrid approach, which combines NLP, grounded theory, and statistical analysis, opens new opportunities for policy evaluation research. This method not only preserves the integrity of policy texts but also improves the efficiency of policy analysis while reducing reliance on human resources. Consequently, this approach can be adapted for the quantitative evaluation of policy texts in other fields.

Third, from a theoretical standpoint, this study provides substantial evidence for refining China's national park policies through a detailed deconstruction of relevant texts. The focus was on a comparative analysis of the similarities and differences between national park policies at the central and local levels, as well as the heterogeneity across the policy texts of the first five national parks. Thus, the insights derived from this investigation may contribute significantly to the literature on national park policy evaluation, both in China and internationally.

## 5.3 Implications

First, utilizing a specially constructed dictionary, this study developed a quantitative evaluation method for national park policy texts. Core vocabulary from China's national park policy discourse was identified and quantified through NLP, resulting in a comprehensive dictionary that encapsulates the essence of these policies. This tool enables policymakers at various levels to clearly understand the focal points of individual policies. Additionally, a quantitative index system comprising seven primary indicators and 49 secondary indicators was established, forming a framework that may facilitate cross-country quantitative comparisons of national park policy texts.

Second, this study provides a critical overview of the main achievements and ongoing challenges in China's current national park policy landscape, laying the foundation for future policy formation, revision, and repeal. The findings indicate that contemporary policies prioritize the establishment of unified management institutions and strict ecological protection, yet often overlook aspects such as traditional cultures, community engagement, and natural disaster mitigation. Future policy development should build on these findings by creating more targeted strategies to preserve cultural heritage, enhance community welfare, and implement scientific disaster prevention and management.

Lastly, this research serves as a window through which the global community can gain insights into the evolution of China's national park system. A systematic review of the central government's top-level design and local government responses highlights China's innovations in ecological protection and livelihood improvement. Initiatives such as "one household, one post (一户一岗)" and "building and sharing (共建共享)" exemplify China's contributions to global ecological conservation efforts, offering distinctly Chinese perspectives on these universal challenges.

## 6. Limitations, future research directions

Our results reveal that: (1)The central government's planning priorities for the national park system focus on ecological conservation and sustainable livelihoods for indigenous communities; (2) Significant variations exist in the implementation of national park policies by local governments; (3) The five inaugural national parks exhibit marked differences in conservation objectives, integrated management practices, spatial planning, and industrial development trajectories.

This paper introduces a novel policy evaluation methodology that integrates NLP, grounded theory, and statistical analysis. While this method can facilitate objective and rational policy evaluation, it still requires further refinement and more comprehensive validation. Additionally, the strategic framework of national park policies at both central and local levels was explored, using policy texts as the foundation. Due to time constraints and limited access to some government

websites, this study was unable to obtain all national park-related policy documents in China. Therefore, the study results may be subject to potential bias.

This work can be further improved in the future by considering three key aspects:

Future efforts should further strengthen comparative research, particularly comparative research on national park policy texts between countries with different political systems or different levels of development, and further explore their differences and underlying causes, thereby providing successful experience for global ecological governance.

The causes of local policy heterogeneity require deeper exploration. While potential causes were identified in this study, a more detailed investigation into the interplay and impacts of these elements on the heterogeneity of local national park policies is necessary.

Ongoing monitoring and analysis of the implementation and actual effects of national park policies should be prioritized. The policy cycle includes stages such as agenda-setting, policy formulation, implementation, and evaluation [62]. Evaluating the success of China's national park policies can be enhanced by considering a broader range of perceptions and outcomes across political, economic, cultural, social, and ecological dimensions from the perspective of local residents.

## Supporting information

**S1 Appendix. Variable settings in the quantitative evaluation system for national park policies in China.** This table presents the hierarchical coding framework developed for the systematic analysis of national park policy documents. The framework is structured around seven primary variables (X1-X7) and 49 secondary indicators. Each primary variable is subdivided into secondary variables, which are further defined through specific, clearly delineated fundamental categories. (DOCX)

**S2 Appendix. Study constitute the proposed framework for quantitative analysis of China's national park policy texts.** This conceptual model visually maps the hypothesized pathways of influence among the seven core policy dimensions identified in the analytical framework (S1 Appendix). The diagram depicts a sequential and interactive logic: Policy Basis stimulates the formulation of Policy Objectives. These objectives are to be achieved through specific Policy Measures, which in turn ensure the implementation of Policy Guarantees. This combined policy apparatus acts within defined Policy Areas and on key Policy Receptors. The ultimate outcomes of this entire process are the realization of various Policy Functions. (PDF)

**S3 Appendix. Average scores and standard deviations of secondary variables between two levels.** This table presents the average scores for 49 secondary policy variables, calculated separately for aggregated central-level and local-level policy documents. The variables span all seven principal dimensions of the analytical framework. For each variable, the table lists its average score at the central level, its average score at the local level, and the standard deviation between these two values. The standard deviation quantifies the degree of divergence in emphasis on each specific policy element between the two administrative tiers. (DOCX)

**S4 Appendix. Samples of master plans for China's first five national parks.** This table provides the reference codes, full names, and release periods for the key national-level policy documents on National Park planning that form the core of the policy analysis. The listed master plans, including those for the Northeast Tiger and Leopard, Sanjiangyuan, Hainan Tropical Rainforest, Wuyishan, and Giant Panda National Parks, represent recent, long-term strategic frameworks and serve as primary sources for the subsequent textual and content analysis. (DOCX)

**S5 Appendix. Secondary variable scores and standard deviations for the five national park policy texts.** This table reports the quantified metrics (e.g., frequency counts or standardized scores) for 49 secondary variables extracted from

the content analysis of five key policy documents (P7, P9, P19, P23, P26). The final column provides the standard deviation for each variable, reflecting the variation in its prominence across the different plans.
(DOCX)

**S6 Data. This data presents the dictionary we obtained after using a Python program to perform word segmentation, exclude non-essential words, and include specialized terms.** Based on this, we further conducted the word-frequency tallying and sorting.
(TXT)

## Author contributions

**Conceptualization:** Jun-Hui Li.

**Data curation:** Hai-Tao Yu.

**Formal analysis:** Jun-Hui Li.

**Funding acquisition:** Yuan Wang.

**Investigation:** Yuan Wang.

**Methodology:** Hai-Tao Yu, Yuan Wang.

**Project administration:** Yuan Wang.

**Resources:** Hai-Tao Yu, Wen-Lie Chen.

**Software:** Hai-Tao Yu, Wen-Lie Chen.

**Supervision:** Wen-Lie Chen.

**Writing – original draft:** Jun-Hui Li.

**Writing – review & editing:** Wen-Lie Chen.

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
