## [Decision Letter · Decision Letter 0]

29 Sep 2025

Dear Dr. Yu,

Thank you for submitting your manuscript to PLOS ONE. After careful consideration, we feel that it has merit but does not fully meet PLOS ONE’s publication criteria as it currently stands. Therefore, we invite you to submit a revised version of the manuscript that addresses the points raised during the review process.

We look forward to receiving your revised manuscript.

Kind regards,

Qiwei Ma

Academic Editor

PLOS ONE

Journal Requirements:

“          This research was funded by the National Social Science Foundation of China, grant number 20XSH022, and by the Philosophy and Social Science Planning key Project of Qinghai Province, grant number 23ZCZD007, and by the Philosophy and Social Science Foundation of Guangxi Zhuang Autonomous Region, grant number 21GMZ009.”

4. Please include captions for your Supporting Information files at the end of your manuscript, and update any in-text citations to match accordingly. Please see our Supporting Information guidelines for more information: http://journals.plos.org/plosone/s/supporting-information .

Reviewers' comments:

Reviewer's Responses to Questions

**Comments to the Author**

1. Is the manuscript technically sound, and do the data support the conclusions?

Reviewer #1: Yes

Reviewer #2: Yes

2. Has the statistical analysis been performed appropriately and rigorously?

Reviewer #1: Yes

Reviewer #2: Yes

3. Have the authors made all data underlying the findings in their manuscript fully available?

Reviewer #1: No

Reviewer #2: Yes

4. Is the manuscript presented in an intelligible fashion and written in standard English?

Reviewer #1: Yes

Reviewer #2: Yes

Reviewer #1: The authors examined central and local management documents related to national parks using a natural language model and the grounded theory approach. The manuscript is well-structured and employs an innovative methodological approach to address the research questions. The text is clear, and the quality of the figures is good. However, I have a few suggestions for improving the content:

Abstract: It would be advisable to begin the abstract not with the presentation of methods, but rather with a general and specific research problem that the study aims to address.

Introduction: The Introduction should provide a more in-depth presentation of China's relationship with national parks and how this has evolved. Currently, some relevant information appears only in Section 5.1 (Discussion), which could also be included in the Introduction.

Hybrid Model Explanation: Closely related to the above point, the hybrid model should be explained in greater detail in the Introduction (e.g., roles and tools at different territorial levels). This would help readers better understand the significance of comparing the top and local levels later in the study.

Research Novelty Emphasis: The authors, in my opinion, emphasize the novelty of the study and the identified policy gap too frequently throughout the manuscript. Instead, the research goals and questions could be elaborated on more thoroughly in the Introduction. It also seems unusual that Section 5.2 separately summarizes how the study contributes to the evaluation of national park management policies. While this topic is certainly worth discussing, it might be more useful to integrate these points into the relevant sections of the text. Additionally, several references to this contribution appear in Section 5.3 (Implications) too.

Table about the meaning of Basic Variables: In Section 3.3.1, a table listing examples related to the 49 basic variables (e.g., policy measures, policy basis) would be helpful. This would clarify their content and ensure a consistent understanding among readers, as the explanatory sentences currently provided in the Results and Discussion sections may be insufficient for full comprehension.

In my opinion, these suggestions help to strengthen the manuscript’s clarity and coherence.

Reviewer #2: The manuscript examines the development and management of China’s national park system through a quantitative evaluation of 77 policy texts issued between 2013 and 2023. Using a mixed-method approach that combines natural language processing, grounded theory, and statistical analysis, the study compares the central government’s top-level design with local government responses and identifies heterogeneity across the first five national parks. The findings highlight both alignment and divergence in policy objectives and practices, offering insights for refining China’s national park governance and contributing to global discussions on conservation policy. Please check for suggestions and recommendations in the attachment.

**Do you want your identity to be public for this peer review?** For information about this choice, including consent withdrawal, please see our Privacy Policy

Reviewer #1: No

Reviewer #2: No

---

## [Author Response · Author response to Decision Letter 1]

18 Dec 2025

Dear editor and reviewers:

On behalf of my co-authors, I am pleased to resubmit our revised manuscript entitled “Research on the Management of the System Construction of National Parks with China Characteristics: Evidence from Policy Texts” for consideration in PLOS ONE. We are very grateful to you and the reviewers for the constructive and detailed feedback on our previous submission. We have carefully revised the manuscript to address all concerns. The key improvements include:

• Abstract: We have shortened certain sentences to enhance clarity. And we have expanded the description of this study's national significance. This study not only contributes to the global understanding of China's national park governance characteristics, but also provides valuable insights for national park development in developing countries and offers lessons for global ecological governance.

• Introduction: We have removed the description (e.g., average park sizes) from the revised manuscript. And we explicitly state that “And no prior studies quantitatively evaluated national park policy texts in China using NLP and grounded theory” and “it is the first quantitative evaluation of Chinese national park policy texts using NLP and grounded theory” in the manuscript. Updated and expanded the citations to reflect the most recent scholarship.

• Literature Review: The existing research findings have been compiled and organized thematically. For national park policy research, the thematic review focuses on three core dimensions: policy instruments, challenges, and objectives. The methodology for policy text analysis has been consolidated through two primary approaches—single-policy evaluation techniques and the PMC index. Concurrently, a critical synthesis of existing studies has been undertaken, identifying shortcomings in prior research while highlighting the innovative contributions of this paper. The bibliography has been updated and expanded to reflect the latest scholarly developments. We believe these revisions significantly enhance the manuscript's clarity, rigour, and scholarly contribution, directly addressing the reviewers' concerns.

• Materials and Methods: We have provided more detailed data collection information: search terms, inclusion/exclusion criteria, and data sources. Independent coding was performed by two experts, with Cohen's kappa coefficient tested. The use of Python software packages was also specified. Furthermore, we have continuously adjusted the formatting of tables, figures, and equations in accordance with established requirements.

• Results: The Discussion Part has been reframed to incorporate an international comparative perspective, a critical reflection on the limitations of text-based analysis, and a deeper interpretation of China-specific findings, thereby significantly enhancing the study's rigour and relevance. At the same time, we have explained and clarified the standard deviation, emphasizing its significance in addressing heterogeneity.

• Discussion, Implications, and Findings: We have revised the discussion and added the novel insights for international audiences. At the same time, comparative studies have been incorporated to clarify the significance of this research for developing nations establishing national parks. Incorporate critical reflection on potential biases into policy text analysis, such as the fact that policy content may not necessarily reflect actual implementation.

• Limitations, Future Research Directions: The conclusion section highlights three key points. Furthermore, critical analysis has been introduced to further highlight the limitations inherent in this study. As the acquisition of policy documents relies on publicly available publications, unpublished policy content has not been included in the study. It should also focus on comparative international research, exploring comparisons between China's national park policies and those of other developing nations to provide richer insights for global ecological governance.

• References: The existing references have been reformatted according to Vancouver style; duplicate entries have been removed; outdated references have been excluded, and the latest references have been added.

We respectfully resubmit this revised version for your kind consideration, and we sincerely hope it now meets the standards for publication in PLOS ONE.

Thank you again for the opportunity to revise and resubmit.

Sincerely,

Hai-Tao Yu

---

## [Editor Report · Decision Letter 1]

29 Dec 2025

Research on the Management of the System Construction of National Parks with China Characteristics: Evidence from Policy Texts

PONE-D-24-46176R1

Dear Dr. Yu,

We’re pleased to inform you that your manuscript has been judged scientifically suitable for publication and will be formally accepted for publication once it meets all outstanding technical requirements.

Kind regards,

Qiwei Ma

Academic Editor

PLOS One
---

## [Editor Report · Acceptance letter]

PONE-D-24-46176R1

PLOS One

Dear Dr. Yu,

I'm pleased to inform you that your manuscript has been deemed suitable for publication in PLOS One. Congratulations! Your manuscript is now being handed over to our production team.

Kind regards,

on behalf of

Prof. Qiwei Ma

Academic Editor

PLOS One